# Apixaban-Induced Esophagitis Dissecans Superficialis-Case Report and Literature Review

**DOI:** 10.3390/diseases12100263

**Published:** 2024-10-21

**Authors:** Alexandru Ionut Coseru, Irina Ciortescu, Roxana Nemteanu, Oana-Bogdana Barboi, Diana-Elena Floria, Radu-Alexandru Vulpoi, Diana Georgiana Strungariu, Sorina Iuliana Ilie, Vadim Rosca, Vasile-Liviu Drug, Alina Plesa

**Affiliations:** 1Gastroenterology and Hepatology Institute, “Saint Spiridon” University Hospital, 700111 Iasi, Romania; ionutz_ionutz_barlad@yahoo.com (A.I.C.); irinaciortescu@yahoo.com (I.C.); oany_leo@yahoo.com (O.-B.B.); iovdiana95@gmail.com (D.-E.F.); vulpoi.radu@yahoo.com (R.-A.V.); dianageorgiana06@yahoo.com (D.G.S.); sorinaabutura@yahoo.com (S.I.I.); vroshca94@gmail.com (V.R.); vasidrug@email.com (V.-L.D.); alinaplesaro@yahoo.com (A.P.); 2Medical I Department, University of Medicine and Pharmacy “Grigore. T. Popa”, 700115 Iasi, Romania

**Keywords:** apixaban, upper gastrointestinal bleeding, esophagitis dissecans superficialis

## Abstract

Novel direct oral anticoagulants (DOACs) are prescribed worldwide in the treatment of non-valvular atrial fibrillation. Adverse reactions have been reported following the use of DOACs. One notable trend in the literature is the growing number of reported cases of esophagitis dissecans superficialis (EDS) generated by DOAC use. We hereby report the case of a 73-year-old woman who presented to the hospital with asthenia, dysphagia, and melena two days prior to admission. The patient had taken apixaban due to non-valvular paroxysmal atrial fibrillation for a few weeks. The biological panel showed moderate anemia with a hemoglobin level of 7.7 g/dL Apixaban-induced EDS was diagnosed by the characteristic endoscopic findings. The patient received treatment with a proton pump inhibitor (pantoprazole) in a double dose. Also, an iron treatment was recommended for a period of six months. The follow-up endoscopy at one month confirmed the healing of the esophageal lesions. The case was discussed with the cardiologist. The first anticoagulant treatment proposed after discharge was a vitamin K antagonist (acenocumarol) but the patient refused this medication and thus it was decided to initiate rivaroxaban. Although DOACs have demonstrated their efficacy in the prevention and treatment of stroke and thromboembolism among the aging demographic, cases of DOAC-induced EDS will continue to pose numerous challenges for physicians worldwide.

## 1. Introduction

In recent years, direct oral anticoagulants (DOACs) have significantly impacted the management of thromboembolic conditions, providing an alternative to conventional anticoagulation options such as vitamin K antagonists (VKAs) [1].

The primary agents are dabigatran, rivaroxaban, apixaban, and edoxaban. These medications are broadly classified into two categories: direct factor Xa inhibitors (which include rivaroxaban, apixaban, and edoxaban) and direct thrombin inhibitors (represented by dabigatran) [2]. While DOACs are generally considered to have a favorable safety profile, the risk of GI bleeding compared to VKAs remains a subject of ongoing debate [3].

Apixaban is a powerful direct-acting oral medication that effectively blocks the function of factor Xa [4]. It is extremely selective and its effects may be reversed if needed [5]. Unlike other medications, it does not rely on antithrombin III to prevent blood clot formation. Apixaban hinders the action of both free and clot-bound factor Xa, as well as prothrombinase activity, hence it impedes the formation of blood clots [6]. Esophagitis dissecans superficialis (EDS) is an uncommon benign esophageal disorder defined by dysphagia and a sloughing of the squamous mucosal lining. When accompanied by a minimal inflammatory response, this can be a diagnosis of exclusion [7]. The condition may be caused by various insults, ranging from autoimmune, mechanical, or chemical agents. In certain cases, patients may exhibit no symptoms or experience dyspepsia, dysphagia, odynophagia, or non-cardiac chest pain. The unique endoscopic appearance usually guides the diagnosis as the “gift-wrap ribbons” are easily detected in the distal esophagus [8].

They embody the desquamation of sheets of superficial squamous mucosa in a vertical manner (i.e., sloughing esophagitis). Treatment may be recommended among symptomatic patients, but the condition usually subsides after removal of the etiological factor. No life-threatening complications have been cited in the literature up to this point, with a generally favorable clinical evolution reported. In relation to symptomatic treatments, the use of proton pump inhibitors (PPIs) seems to have beneficial effects [9]. New generation DOACs have been linked to changes in the integrity of the esophageal mucosa, yet there is still a scarcity of data regarding the role of apixaban in prompting EDS [10]. We hereby report the case of a female patient who developed EDS shortly after initiating treatment with apixaban.

## 2. Case Presentation

A 73-year-old female patient was referred to our Gastroenterology Department presenting with asthenia, mild dysphagia, and melena noted two days prior to admission. Upon examination, she exhibited normal vital signs, skin pallor, a soft abdomen, and arrhythmic heart sounds. The digital rectal examination confirmed melena. She had been prescribed 5 mg of apixaban twice daily for the past month in the context of non-valvular paroxysmal atrial fibrillation. The patient was not taking any other medications known to have adverse gastrointestinal (GI) effects, such as nonsteroidal anti-inflammatory drugs, bisphosphonates, sunitinib, or antibiotics. Additionally, she reported no history of smoking or alcohol use. Several days after initiating therapy with apixaban, the patient complained of some discomfort while eating and even struggled to pass food down, but her symptoms were alleviated by consuming smaller meals.

The blood work revealed moderate anemia with a hemoglobin level of 7.7 g/dL (normal range 12–16 g/dL), an erythrocyte count of 2.0 M/mL (normal range (NR) 4.2–5.5 M/mL), a mean corpuscular volume of 68 fL(NR 78–95 fL), a mean corpuscular hemoglobin of 22 g/dL (NR 32–36 g/dL), and a platelet count of 175.000/mmc (NR 150,000–450,000/mmc). The biochemistry showed an iron level of 40 mcg/dL (NR 38–140 mcg/dL), with a ferritine level of 17 mcg/L (NR 10–280 mcg/L). The kidney (creatinine level = 0.63 mg/dL with NR 0.55–1.20 mg/dL; urea level = 19 mg/dL with NR 15–43 mg/dL; and the clearance creatinine using the Cockcroft–Gault equation was 94 mL/min) and the hepatic function were normal (alanine transaminase = 17 U/L with NR 10–50 U/L; aspartate aminotransferase = 22 U/L with NR 14–49 U/L; total bilirubin = 0.55 mg/dL with NR 0.40–1.40 mg/dL; direct bilirubin = 0.39 mg/dL with NR 0.39–1.40 mg/dL; the gamma-glutamyl transferase = 35 U/L with NR 15–50 U/L; the alkaline phosphatase = 30 U/L with 16–48 U/L; and the lactate dehydrogenase = 200 U/L with NR 140–280 U/L). The coagulation tests were within normal ranges (international normalized ratio = 1.09 with NR 0.8–1.1; fibrinogen level = 206 g/L with NR 200–400 g/L; and the prothrombin time = 13 s with NR 11–13.5 s).

She underwent an upper GI endoscopy which revealed pinkish longitudinal mucosal strips adhering to the middle and lower esophagus, with mucosal erosions observed following detachment, and normal looking mucosa between the lesions. The endoscopic appearance of the esophageal casts, also known as gift-wrap ribbons or wrap paper, was suggestive of EDS or sloughing esophagitis (Figure 1). The stomach mucosa appeared to be normal.

No bleeding was detected in the upper GI tract at the time of the endoscopic examination. The histological assessment of the biopsy samples noted parakeratosis and leucocyte infiltration. Considering the endoscopic findings, the clinical presentation, and the patient history, a diagnosis of EDS was established. A Colonoscopy was also performed, which did not find any lesions. Etiological factors such as gastroesophageal reflux disease, human immunodeficiency virus, and herpes simplex infection were ruled out by specific tests.

Discussing the case with the patient’s cardiologist, apixaban was discontinued and a treatment with low-molecular-weight heparin twice daily was initiated during her hospital stay. After discharge, a treatment with rivaroxaban was prescribed. She also underwent acid suppression with 40 mg of a PPI (pantoprazole) twice daily and a treatment with an oral iron supplement for six months. A liquid diet was initiated for 24 h. Additionally, she was advised to ingest these medicines with enough water and stay seated for at least 30 min after taking any drugs to prevent them from adhering to the esophageal mucosa. A follow-up upper GI endoscopy at one month revealed a significantly improved esophageal appearance (Figure 2).

## 3. Discussion

Esophagitis dissecans superficialis, also known as desquamative esophagitis or exfoliative esophagitis, is a rare entity in clinical practice, and was first reported in 1892 by Rosenberg [11]. To date, the incidence rate is estimated at 0.03% of cases, as reported by Akhondi et al. in a large study covering over 20,000 upper digestive endoscopies [12]. Numerous other reports over the years have shown that this condition is infrequent, and its discovery is usually accidental [13].

Most cases typically present with mild symptoms, such as heartburn and epigastric pain. Medications such as dabigatran and sunitinib, exposure to corrosive substances, autoimmune disorders (bullous pemphigoid), caustic ingestion, chemical irritants, and candidiasis are conditions linked to its occurrence [14]. With regard to the case presented, DOACs may be responsible for EDS. The explanation resides in the chemical composition of DOACs, for example, dabigatran has a tartaric acid core that is coated by dabigatran etexilate, which is directly responsible for the esophageal injury. Tartaric acid is not included in the composition of apixaban [15]. However, the coating of apixaban contains sodium lauryl sulfate [16], which is a surfactant known to cause irritation and desquamation of human mucous membranes such as the mucosal layer [17].

The pathophysiological mechanisms that lead to esophageal injury are currently not well defined, although they likely arise from either ischemic events or direct damage to the mucosal layer. Endoscopic inspection and collection of histological samples can highlight an esophageal mucosal injury, with three macroscopic criteria that are strongly associated with this diagnosis [18].

The histological features of this condition are as follows: splitting within the epithelium, significant parakeratosis, degeneration of cysts within the epithelium, and an increase in basal cells [19]. A variety of patients are affected by EDS, which is a condition that has a distinctive endoscopic appearance. Endoscopic features of EDS may include single or multiple white regions of peeling mucosa that extend from the middle to the distal esophagus, or even a diffuse sloughing of the entire esophageal mucosa.

Three endoscopic diagnostic criteria have been suggested: sloughed mucosa segments that are at least 2 cm in length, normal underlying mucosa, and non-ulcerated surrounding mucosa [20].

The article by Prasoppokakorn et al. [21] published in 2019 included a summary of the existing literature with 30 cases of EDS. We hereby highlight the most important findings:Out of the 30 patients undergoing endoscopy, 21 (70.0%) patients had characteristic features of EDS;The histology of esophageal biopsies found a sloughing of the superficial layer of epithelium in all cases, highlighted the presence of eosinophils on the surface in 13.3% of patients, and observed parakeratosis in 56.7% of cases, while 6.7% had normal underlying mucosa;All 30 patients had a favorable clinical outcome, with improvements in endoscopic appearance irrespective of acid suppressive therapy with PPIs (30% without PPI treatment, 70% with PPI treatment).

Currently, no established diagnostic or therapeutic algorithm for EDS exists. To summarize the existing relevant data concerning EDS, we performed a literature search for all additional case reports and case series of EDS published following the article by Prasoppokakorn T et al. (between August 2019 and July 2024) (Table 1) [20,22,23,24,25,26,27,28,29,30].

Most patients experience a favorable outcome with EDS, despite its occasional dramatic presentation. Acid suppression, topical analgesics, and the cessation of any potential precipitating medications typically lead to the healing of EDS without any sequelae. Stricture development is believed to be uncommon due to the superficial character of the epithelial injury in this disease [27,28]. Our findings indicate that the use of PPIs is effective, and that a biopsy is essential for ruling out other illnesses.

Establishing an accurate diagnosis of EDS is challenging due to the presence of esophageal mucosa lesions in a variety of diseases, most of which lack unique features. A proper diagnosis involves a thorough examination of both the macroscopic and microscopic features, as well as a thorough review of the patient’s medical history and the risk factors mentioned in the current literature [29]. This case has particular significance since it is the first known case of a patient experiencing upper GI bleeding as the first sign of EDS, while using apixaban which is known for its strong safety profile. Apixaban is increasingly being used worldwide as a therapy for atrial fibrillation, following the risk scores and indications of guidelines [30].

The type of anticoagulant may impact the risk of substantial upper GI bleeding, which is a common and possibly severe consequence of oral anticoagulant medications [35]. A recent study estimated that the incidence rate for upper GI bleeding in patients receiving DOACs is 20.9%. Interestingly, apixaban was found to carry the lowest risk of developing upper GI bleeding [36]. Apixaban appears to be the safest type of DOAC in terms of hemorrhagic adverse events, as reported by Ray et al. [37].

The first treatment of choice after discharge was a vitamin K antagonist (VKA) (acenocumarol), but she refused because she did not want to have her INR monitored at regular intervals. Apixaban was considered as a potential trigger factor for EDS so it was discontinued. Edoxaban and warfarin were not available in our country at this time. Dabigatran, according to specialized literature, presents a risk of EDS, so it was decided that is was not a valuable option in our case [19,28]. Rivaroxaban was chosen as the new DOAC after consultation with the cardiologist and taking into account the patient’s decision regarding the use of the VKA (acenocumarol).

We proposed the next diagnostic and treatment approach (Figure 3).

A study conducted by Purdy et al. concluded that the lesions are usually located as follows: 42% in the distal esophagus, 23% in the entire esophagus, 19% in the middle esophagus, and 8% in the proximal esophagus [9].

A thorough examination of both the macroscopic and microscopic features is essential, as well as a review of the patient’s medical history and potential risk factors [38]. We considered drug–drug interactions in our scenario. The patient received statins which could interfere with the metabolism of permeability glycoproteins, and hence might raise the risk exposure for apixaban [39]. However, the available research data validate the use of statins as a safe medicine in combination with apixaban for preventing significant thromboembolic events in patients with non-valvular atrial fibrillation [40].

This case represents the first documented instance of upper GI bleeding as the initial manifestation of EDS in a patient receiving apixaban. This occurrence is particularly significant considering the previously established favorable safety profile of apixaban. The novelty of this clinical case lies in the reporting of a potential additional drug that can cause EDS and subsequent upper GI bleeding. The prognosis is excellent.

Moreover, we propose a diagnostic and therapeutic algorithm based on data from the most relevant 40 case reports and case series in the literature.

The occurrence of melena was ascribed to the administration of apixaban, and it resolved following cessation. Pantoprazole was administered and complete resolution of the symptoms was obtained. At this moment, there are no clear guidelines regarding the use of a double-dose IPP in EDS, so we decided to treat the patient as having severe esophagitis according to our hospital protocol [41]. We suspected apixaban was the triggering factor.

## 4. Conclusions

This case represents the first documented instance of upper GI bleeding as the initial manifestation of EDS in a patient receiving apixaban. Choosing a treatment for EDS was difficult, in the absence of clear guidelines. Also, we encountered difficulties in choosing the right anticoagulant treatment for our patient, for both objective and subjective reasons (the patient’s decision).

## Figures and Tables

**Figure 1 diseases-12-00263-f001:**
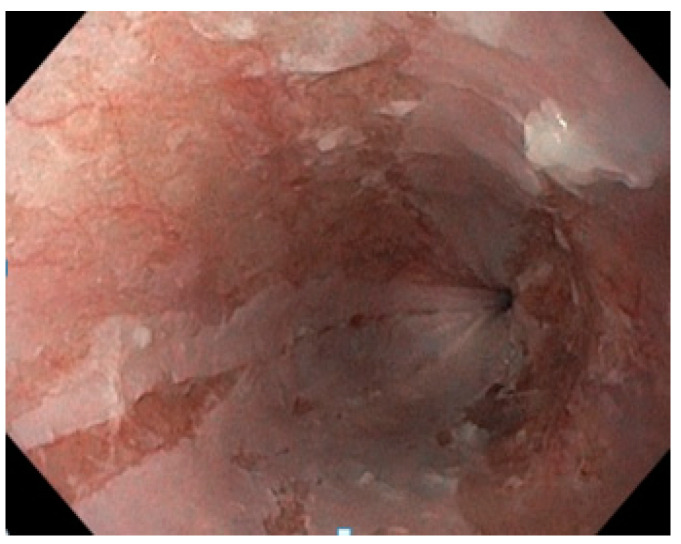
Endoscopic findings showing the esophageal mucosa with longitudinal strips from the sloughing of the mucosa with a “gift-wrap ribbons” or “crepe-paper” appearance.

**Figure 2 diseases-12-00263-f002:**
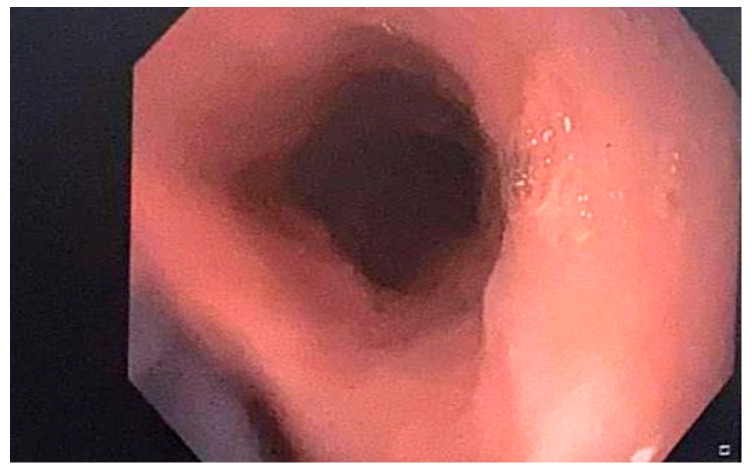
Improved endoscopic appearance of the esophageal mucosa one month after treatment.

**Figure 3 diseases-12-00263-f003:**
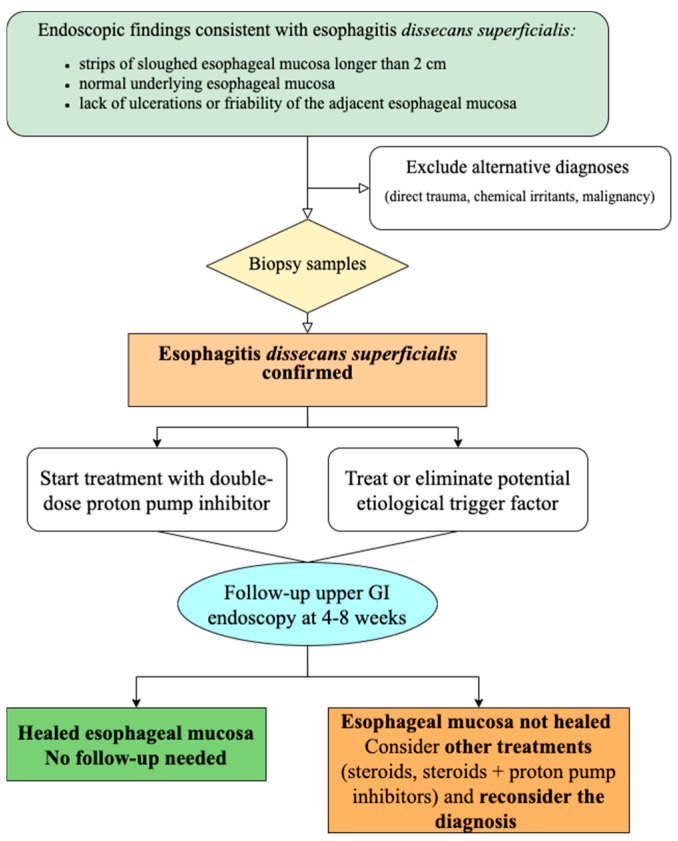
Proposed diagnostic and therapeutic algorithm in patients with esophagitis dissecans superficialis.

**Table 1 diseases-12-00263-t001:** Summary table detailing existing case reports and case series on esophagitis dissecans superficialis, highlighting the suspected etiological factors and clinical management.

Endoscopic Follow-Up	Treatment	Biopsy	Etiology	Patient	Author, Year, [Ref.]
Follow-up endoscopy at 1 week (healed)	Liquid diet, iron supplementation was withheld	Yes	Medication-induced esophagitis (oral iron supplementation)	94-year-old male	Nasir UM et al., 2020 [31]
Follow-up endoscopy at 1 year (healed)	High-dose proton pump inhibitor	Yes	COVID-19 infection	68-year-old female	Salehi AM et al., 2022 [32]
None	Oral diclofenac was discontinued, treated with oral sucralfate and high-dose proton pump inhibitor	Yes	Medication-induced esophagitis (oral diclofenac)	54-year-old male	Senyondo G et al., 2022 [33]
None	Oral prednisone	Yes	Autoimmune disease (bullous pemphigoid)	67-year-old male	Morel-Cerda EC et al., 2020 [34]
None	Double-dose proton pump inhibitor and sucralfate	Yes	Treatment with tyrosine kinase inhibitor	70-year-old male	Kanagalingam G et al., 2021 [22]
None	Oral proton pump inhibitor and sucralfate	No	Treatment with metotrexate	57-year-old male	Venkata R et al., 2020 [23]
Follow-up endoscopy at 1 week (healed)	Intravenous proton pump inhibitor	No	Following hematopoietic stem cell transplantation	18-year-old female, 29-year-old male, 21-year-old male, 16-year-old male	Iwamuro M et al., 2020 [24]
None	Proton pump inhibitor treatment	Yes	Treatment with dabigatran	77-year-old women	Zhou Y et al., 2020 [25]
Follow-up endoscopy at 1 week (healed)	Double-dose proton pump inhibitor	Yes	Treatment with sunitinib	66-year-old male	Gayam S et al., 2016 [20]
Yes (healed)	Steroid treatment	Yes	Dermatomyositis	36-year-old male	Zheng W et al., 2021 [26]

## Data Availability

Data are contained within the article.

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
