# Peer review of "Apixaban-Induced Esophagitis Dissecans Superficialis-Case Report and Literature Review"

_diseases, 2024, doi:10.3390/diseases12100263_

Round 1

Reviewer 1 Report

Comments and Suggestions for Authors

Despite the complexity of guidelines, physicians often face challenges in selecting the appropriate anticoagulant agent, especially in patients with multiple comorbidities. It is crucial to note that apixaban, a commonly used anticoagulant, has been associated with ESD. Therefore, it is imperative to provide patients with clear instructions on the proper administration of this medication. Authors, the first documented instance of upper GI bleeding as the initial manifestation of EDS in a patient receiving apixaban underscores the need for caution. This occurrence, in the context of the previously established favorable safety profile of apixaban, calls for hopeful and optimistic approaches. Furthermore, acid suppression with double-dose PPI may offer a promising solution.

The authors have clearly presented the issues they are studying in the manuscript with photographs that confirm their discussion.

Suggested minor correction:

In the introductory part, the authors could write a little more (several sentences) about novel direct oral anticoagulants (DOACs), which groups these drugs structurally belong to.

Authors could write a few sentences about apixaban: directly inhibiting factor X, pharmacokinetics. Is apixaban special in any way from DOACs?

Reviewer 2 Report

Comments and Suggestions for Authors

The Romanian authors reported on a 73-year-old woman with possibly apixaban-induced esophagitis dissecans superficialis (EDS) causing dysphagia and melena. Although it is an informative case report, I have some concerns as follows.

1.       In Abstract, redundant descriptions regarding direct oral anticoagulants (DOACs) are unnecessary. The direct information of this case is recommended to be described there, because this is a case report.

2.       Although it is noted “The biochemistry panel was unremarkable”, I wonder whether the patient was unassociated with an iron deficiency.

3.       It is noted “Etiological factors such as gastroesophageal reflux disease, human immunodeficiency virus and herpes simplex infection were ruled out by specific tests”. I would strongly recommend the authors to show actual data in the specific tests they seemed to have conducted.

4.       Although it is noted “She also underwent acid suppression with PPIs 40 mg twice daily”, the actual chemical name of the PPI they used was not described in Case presentation. If it was pantoprazole as described in Discussion, it is necessary to describe so in the part of Case presentation.

5.       It is necessary to describe the reason why rivaroxaban was chosen as a DOAC after discharge.

6.       The part of Discussion is redundant too much. I would recommend the authors to describe contents closely related to this case and a short review, because this is mainly a case report just with literature review.

7.       I wonder whether “Proposed diagnostic and therapeutic algorithm in patients with EDS” shown in Figure 3 could be validated just based on this case.

8.       I wonder why the words “Apixaban” and “apixaban” were used mixedly in the main text. I would recommend the authors to receive an appropriate revision by a professional or experienced editor for the composition and language.

Comments on the Quality of English Language

Nil

Round 2

Reviewer 2 Report

Comments and Suggestions for Authors

Although the authors had attempted to improve their manuscript, the revision they made was not enough. My concerns are as follows.

1.       In Abstract, the sentences “The emergence of novel direct oral anticoagulants (DOACs) has significantly transformed the approach to managing elderly patients, improving effectiveness in controlling thrombotic events associated with atrial fibrillation. . The utilization of DOACs offers various benefits with 16 fewer interactions compared to vitamin K antagonists.” are unnecessary. More details of this case should be added there instead.

2.       The yellow marked sentences in the beginning of Introduction should be shortened.

3.       Although it is noted “The rest of biological panel was unremarkable, with normal kidney and liver function, and coagulation parameters within normal limits.”, the results of kidney and liver function tests and coagulation parameters are better to be added, which should have been kept in the patient chart.

4.       It is mandatory to explain why rivaroxaban was chosen as a safer medicine than dabigatran and warfarin for non-valvular paroxysmal atrial fibrillation in a 73-year-old woman.

5.       It is necessary to show clear evidence for the necessity of double-dose PPI in the beginning of EDS treatment.

6.       Descriptions in Conclusions are too subjective and redundant. The conclusive message the authors could show is just what they experienced in this case.

Comments on the Quality of English Language

Nil

Round 3

Reviewer 2 Report

Comments and Suggestions for Authors

I appreciate the authors' efforts to have revised twice this case report.